# Healthcare Transition in Inherited Metabolic Disorders—Is a Collaborative Approach between US and European Centers Possible?

**DOI:** 10.3390/jcm11195805

**Published:** 2022-09-30

**Authors:** Jessica I. Gold, Karolina M. Stepien

**Affiliations:** 1Division of Human Genetics, Children’s Hospital of Philadelphia, Philadelphia, PA 19104, USA; 2Inherited Metabolic Disorders Department, Salford Royal NHS Foundation Trust, Salford H6 8HD, UK; 3Division of Diabetes, Endocrinology and Gastroenterology, University of Manchester, Manchester M13 9PL, UK

**Keywords:** transition service, adult metabolic medicine, collaboration, challenges, inherited metabolic diseases, healthcare systems

## Abstract

Inherited metabolic diseases (IMDs) are rare heterogenous genetic conditions. Advanced technology and novel therapeutic developments have led to the improved life expectancy of patients with IMDs. Long-term, they require close surveillance from specialist adult metabolic providers. Healthcare transition (HCT) is the planned, purposeful process of preparing adolescents for adult-centered medical care and has been recognized globally as a necessary component of care for IMDs. Two recent surveys outlined barriers to the HCT in the US and the UK. The limited knowledge of IMDs among adult physicians was one of the barriers. Some work on specialty curriculum has started and aims to improve the structured training and awareness of rare diseases. Other barriers included social and legal aspects of adulthood, social, vocational and educational support for young adults, care fragmentation and insurance coverage. Although various HCT tools are available, they cannot always be standardized for IMDs. Despite the remarkable differences in the healthcare systems and physicians’ training, collaboration among metabolic centers is possible. International rare disease alliance may enhance the patients’ management via guidelines development and standardized training for adult metabolic providers.

## 1. Introduction

Inherited Metabolic Diseases (IMDs) encompass an expanding group of rare diseases caused by inherited defects in various biochemical pathways [1]. Although the individual incidence is low (from 1 in 10,000 to 1 in 1 million), the overall incidence of all IMDs ranges from 1 in 800 to 1 in 2500 newborns [1,2].

Recent advances in screening, diagnosis, and management of IMDs have resulted in improved clinical and patient-reported outcomes, increased life expectancy, and new recognition of adult-onset phenotypes [3,4]. Latest estimates suggest that over 90% of patients with IMD now survive past 20 years old and that adults (16–80 years) comprise nearly 50% of those with IMD [3,4,5]. Despite this rapidly growing population, adult phenotypes remain underdefined and often diverge from pediatric ones. Specific challenges exist for metabolic providers—historically, this field was entirely within the purview of pediatricians and adult physicians have limited familiarity with these diagnoses or their management. Given the progressive and complex nature of many IMDs, adult age-related health problems, and multi-specialty support systems, these young adults require long-term surveillance of adult metabolic providers.

Healthcare transition (HCT)—the planned, purposeful process of preparing adolescents for adult-centered medical care—has been recognized globally as a necessary component of care for IMD [1,6,7]. HCT for youths with IMD must recognize these unique barriers to optimize adult well-being. This report compares and contrasts barriers to HCT in two different healthcare systems.

## 2. Report

Two recent surveys answer the growing call to define HCT practice among metabolic providers in the United States (US) and European Union (EU) [1,8]. Both surveys assessed current facilitators and barriers to HCT, identifying several similarities among the US and EU responses. HCT is universally recognized as a critical component of IMD healthcare, yet few providers use validated tools to assess transition readiness. A significant barrier globally is the lack of knowledgeable adult providers in all areas of medicine, especially metabolic and psychology/psychiatry, leading to fragmented care. Limited training opportunities in adolescent and adult metabolic medicine are acknowledged by both groups. Despite the perceived importance of HCT, greater efforts need to be directed toward educating providers and standardizing transition practices.

How can metabolic providers in the US and EU collaborate on IMD-specific HCT tools? At first glance, bridging the wide gap between the nationalized health systems of the EU and the disjointed US amalgam of private and governmental health services seems challenging. However, similar initiatives for creation and dissemination of standardized HCT instruments are needed on both sides of the Atlantic. As an example, the ‘Ready, Steady, Go’ document has been translated into several languages and could be potentially adopted by many countries [9].

Lack of specific adult services or healthcare providers trained to care for rare metabolic diseases results in poor HCT transition program [10] Subspecialty providers are increasingly hailed as critical to HCT efforts due to their sphere of knowledge and strong longitudinal family relationships [11]. These initial steps are medical institution-independent and can be enhanced by integration of global methods. To identify the strengths in our differences, we visit several aspects of metabolic care and HCT in the US and the UK.

A common thread through both surveys was the predominance of pediatrically trained metabolic providers and general challenges with physician recruitment. In the US, metabolic physicians first complete a genetics residency coupled with a more general residency. In total, 67.2% of geneticists in the US are dual-certified in pediatrics and genetics while only 11.4% are dual certified in internal medical/family medicine and genetics [12]. An optional year of accredited training in medical biochemical genetics is gaining popularity and is increasingly required for employment. In total, most metabolic providers receive 4–5 years of post-graduate training in the field of genetics and metabolism.

The UK has a specific pathway for adult metabolic medicine through their Royal College of Pathologists. Physicians receive training in internal medicine or surgery followed by training in Chemical Pathology and Metabolic Medicine. Together, metabolic training in the UK occurs over 10 years. New diagnostic and therapeutic developments in the field of rare diseases has increased an interest among clinicians, but the current training pathway with difficult FRCPath examinations had a negative impact on recruitment. A new accredited curriculum on acute management of IMD is under development. It aims to attract clinicians from different specialties, e.g., neurology, renal medicine or cardiology to develop a special interest in this field [13,14]. Given the diversity of the specialty training backgrounds of clinicians in adult metabolic medicine, the training curriculum requires some flexibility to meet their training needs [15]. In addition, there is overlap of the knowledge-based competencies between the UK adult metabolic medicine training curriculum [13,15] and pediatric biochemical genetics [15].

Nearly 98.9% of US respondents take care of pediatric and adult patients compared to 84.1% of European centers [1,8]. European centers, mainly the UK, are more likely to have a separate adult metabolic clinic, which may vary in clinical scope from only lysosomal storage disorders to only intermediary disorders of metabolism. In contrast, US centers, where adults predominantly receive care in pediatric clinics, tend to offer wide-ranging IMD care, engaging a multidisciplinary team of dietitians, social workers and genetic counselors. Continuity of care at pediatric metabolic clinics likely improves patient adherence to follow-up. The implementation of separate adult metabolic medicine clinics in the UK achieved similar goals. Prior to 2005, all patients were discharged from pediatric IMD clinics at 16 years old. The creation of adult-specific IMD clinics introduced a new medical home for these adolescents, ensuring continual care. It resulted in many childhood-onset cases being re-referred to adult services after many years of no follow up, no treatment or special diets.

Care fragmentation occurs in both health systems. The UK’s National Health Service (NHS) provides government-funding (through taxation) for consultation, investigations, and follow-up appointments. There is little involvement from the private sector. However, while pediatric and adult patients are seen within the same health system, visits occur at different hospitals with separate non-integrated electronic health records. The US system of private and government-funded health insurance leads to greater splintering of healthcare delivery. Many tertiary pediatric hospitals are stand-alone institutions, unaffiliated with any local adult institution. IMD patients who received centralized pediatric medical care may be forced to seek care at several different adult institutions due to limitations of their medical insurance. Adult hospitals that are not affiliated with pediatric institutions rarely have genetic or metabolic specialists. They are unlikely to carry metabolic formula, medications for acute IMD care, or run IMD-specific laboratories [8]. For many American institutions, this prevents adults from receiving acute metabolic care in adult hospitals.

US health insurance coverage changes as patients age. Adolescents must apply for adult Medicaid, the public aid-based insurance, even if previously receiving pediatric Medicaid. Private health insurance is usually employment-based, though 2010’s Affordable Care Act allows youth to remain on their parent’s health insurance through age 26 and created marketplaces to purchase independent plans. Regardless of insurance, many states do not mandate coverage of metabolic formula, medical food, or supplements for adults [16]. In contrast, every employed patient in the UK pays monthly into the National Insurance Scheme, which covers universal healthcare for everyone. Unemployed patients are not required to make these monthly payments, but still receive free medical care. Refugees and certain other citizen groups are also treated for free with access to a General Practitioner, acute medical care and social services support.

Optimal HCT should prepare youth for social and legal aspects of adulthood. The age of majority is 16 in the UK and 18 in the US. At this age, adolescents automatically become their own legal decision-maker, regardless of comorbidities. Youth should receive counseling on decision-making supports, such as guardianship or healthcare power of attorney, prior to this birthday. Guardianship, the legal process to become the primary decision-maker for a young adult, is very complex in the UK. While youths reach adulthood at 16, they do not receive all legal rights until 18 years of age. Thus, a guardian cannot be assigned before 18 years old. Between the ages of 16–18, there is a void for a designated decision-maker, which is especially apparent for patients lacking capacity [17]. For complex options, court involvement may be required [17]. Discussing capacity prior to age 16–18 is crucial for avoiding confusion during medical emergencies or to protect patients from predatory behavior.

Education, vocation, and income supports factor largely into HCT planning. Many youths receive healthcare support, such as physical therapy, through their school system. At the completion of schooling (in the US at 21 and the UK at 18), these supports are discontinued, forcing patients to pay out of pocket or through insurance. Day programs or vocational programs exist for those over 21 but can be expensive and usually have limited availability. Income support in the UK is administered at the national level. For many youths with IMD, their caregivers are also able to register for benefits, allowing them to be compensated for full-time care. In the US, income support is administered at the state level with each state setting their own application criteria. Some states will use a medical diagnosis for approval, while others require documented intellectual disability [18]. This inequity can be harmful for young adults with IMD who may have complex medical needs but normal IQs.

## 3. Discussion

Despite systemic differences, the basic tenets of HCT permit international collaboration. Emphasizing HCT research and training is timely—expanded newborn screening via tandem mass spectroscopy began nearly twenty years ago in many countries [19], leading to a growing population of adolescents with IMD requiring transition to adult-centered medical care. International collaboration has been critical in other HCT-related fields, such as long-term follow-up for survivors of childhood cancer to determine rare late sequalae and compile surveillance recommendations [20]. Rare disease alliances, including the Europe Union’s International Rare Disease Research Consortium (IRDiRC) and the US’s National Organization for Rare Disorders (NORD), illustrate the greater impact of collective research efforts and data sharing [21]. The main HCT principles are consistent throughout these medical systems. Adolescents need preparation for adult-centered medical care, education and vocational opportunities, income support, and assistance with decision-making. Defining measurable outcomes for young adults with IMD is also necessary, permitting evaluation and refinement of HCT programs [5,7]. Some parameters will vary by country—such as insurance lapses or care fragmentation. However, creating unified measures to track and assess HCT internationally is beneficial for all. Here, collaboration between international IMD societies on HCT is a necessary first step. Since these disorders are rare and there is a limited pool of metabolic specialists, international collaboration is necessary to implement full-scale HCT planning including best practice recommendations.

One imperative for designing international IMD-related instruments is that current standardized HCT tools, such as “Ready, Steady, Go” are not comprehensive enough for IMD and IMD-specific assessments have not been validated [22]. Managing a rare disease is challenging in adult-centered healthcare. HCT requires a partnership between pediatric and adult providers. Yet, commitment to HCT is skewed toward greater participation and knowledge from pediatric providers and greater disinterest from adult providers. Many adult clinicians have no knowledge of the clinical care of IMD [23]. Adults are likely to see several care teams due to the multisystemic complications. For patients with intermediary disorder of metabolism, self-management is complex and there is an underlying risk of presenting for acute care in a state of transient encephalopathy [24]. Adults with IMD commonly need to educate their own physicians and act as strong advocates for their care. Providing young adults with the educational tools to accomplish this level of self-care is important [5,7]. Disease-specific guidelines will be necessary to properly inform adult-trained clinicians of current management, long-term complications, and surveillance recommendations. A collaborative approach to develop and disseminate these tools would be beneficial for the international IMD community.

Globally, recruitment of adult metabolic physicians and engagement of local clinicians about IMD are critical barriers to successful HCT and transfer. A stronger international alliance of professional societies and patient support groups could lead to greater promotion of adult metabolic medicine and resource curation. Together, IMD societies could partner with general internal medicine or adolescent medicine organizations to increase IMD representation, for example—by expanding the SSIEM Adult IMD course [15]. As both societies have active adult IMD sections, joint meetings are possible. Additionally, institution-based transition coordinators are instrumental in preparing young adults for this process [7]. Employing a transition coordinator may alleviate HCT barriers due to workforce limitations and clinician knowledge and has been shown to improve health outcomes [25,26]. International IMD societies should lead in developing guidelines that include a job description for transition coordinators. Representatives from many countries should be solicited to collaborate on HCT. The current report is limited by the comparison of two developed countries with individualistic attitudes to health care. Our HCT experience may not be generalizable to nations with more community-based health care.

Last, we must reiterate the difference between healthcare transition and healthcare transfer. HCT is the process of preparing adolescents for medical, social, and fiscal autonomy. Transfer, when a young adult leaves a pediatric practice and initiates care with an adult practice occurs more commonly in the EU (90%) than the US (25%) [1,8]. HCT is required for all adolescents with IMD, even those who will receive from the same IMD provider through adulthood. Metabolic physicians are well poised to promote HCT. International collaboration on HCT will strengthen its delivery.

## 4. Conclusions

Despite the differences in the health systems and physicians’ training, collaboration between metabolic centers is possible. International rare disease alliance may enhance the patients’ management via guidelines development and standardized training for adult metabolic providers. Further research and careful planning of coordination of transition care is required to ensure a smooth patients’ transfer to adult metabolic services and empower them in decision making and improve their adherence to follow-up. The successful transfer of care will result in better patients’ engagement with healthcare system.

## Data Availability

Not applicable.

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
