# Peer review of "Healthcare Transition in Inherited Metabolic Disorders—Is a Collaborative Approach between US and European Centers Possible?"

_jcm, 2022, doi:10.3390/jcm11195805_

Round 1

Reviewer 1 Report

The manuscript titled "Healthcare transition in Inherited Metabolic Disorders – Is a collaborative approach between US and European centers possible?" reviews the clinical transition of care for patients with inherited metabolic disorders.  The authors do a good job of introducing the topic and why it is important. Then they address the current state of the field and reviewed recent surveys performed in the US and EU and the opportunities for international collaboration.

Major

I believe this article would be greatly improved with a discussion section.

           What is the call to action?

How have other international collaborations worked? An example

Julkowska, D., Austin, C., Cutillo, C. et al. The importance of international collaboration for rare diseases research: a European perspective. Gene Ther 24, 562–571 (2017). https://doi.org/10.1038/gt.2017.29

What would be the minimum needed to begin these type of collaborations?

Are there aids that could be adopted such as the mentioned ‘Ready, Steady, Go’ or new aids?

Is there an institution that has a system that could be modelled?

What are the next steps to resolve this problem?

How do we train more doctors for this transition? Is there an organization that would be ideal to partner with?

Training takes time what can be done in the interim to address the lack of trained personnel.

Pediatric training is well understood and accepted. What steps need to be taken to get adolescent and adult IMD training?

Some of this is shortly discussed in the report, but could better organized and expanded in the discussion. 

Minor

The conclusions header should be 3. Based on the headers currently in the manuscript.

The last paragraph in the report section is indented when the rest are not.

Author Response

The manuscript titled "Healthcare transition in Inherited Metabolic Disorders – Is a collaborative approach between US and European centers possible?" reviews the clinical transition of care for patients with inherited metabolic disorders.  The authors do a good job of introducing the topic and why it is important. Then they address the current state of the field and reviewed recent surveys performed in the US and EU and the opportunities for international collaboration.

 Thank you for reviewing our manuscript and providing the suggestions below.

Major

I believe this article would be greatly improved with a discussion section. 

A discussion section has been added to address the comments below. Our discussion section includes: why HCT is a timely priority, how international collaborations have been helpful in rare disease and in HCT, and the role of current clinical societies.

           What is the call to action?

How have other international collaborations worked? An example 

Julkowska, D., Austin, C., Cutillo, C. et al. The importance of international collaboration for rare diseases research: a European perspective. Gene Ther 24, 562–571 (2017). https://doi.org/10.1038/gt.2017.29

What would be the minimum needed to begin these type of collaborations?

We have added several suggestions for ways the international IMD community can initiate collaborations, including greater interaction between the adult working groups of the SIMD and the SSIEM

Are there aids that could be adopted such as the mentioned ‘Ready, Steady, Go’ or new aids?

While the “Ready, Steady, Go” instruments are well designed, they do not have enough detail for the complexity of IMD management, especially for individuals who receive dietary therapy or enzyme replacement therapy. Currently available IMD-specific instruments, such as those on the New England Consortium of Metabolic Providers, have not been validated for use. Therefore, new instruments are needed. We have added these details to our discussion section.

Is there an institution that has a system that could be modelled?

Unfortunately, both the European and the US survey demonstrated a lack of standardization in the HCT process globally. We are unable to highlight a specific institution at this time.

What are the next steps to resolve this problem?

How do we train more doctors for this transition? Is there an organization that would be ideal to partner with? 

Training takes time what can be done in the interim to address the lack of trained personnel. 

Pediatric training is well understood and accepted. What steps need to be taken to get adolescent and adult IMD training?

Some of this is shortly discussed in the report, but could better organized and expanded in the discussion. 

These questions are now addressed in the new discussion section

Minor 

The conclusions header should be 3. Based on the headers currently in the manuscript.

This has been amended. The conclusion section is now 4 due to the addition of a discussion section.

The last paragraph in the report section is indented when the rest are not.

This has been amended.

Reviewer 2 Report

I read with great interest your manuscript on an important topic of healthcare transition (HCT) in IEM which should be planned, standardized, periodically evaluated and of great importance, and as you showed independent of the organization and financing of the health care system, based on international guidelines.  In my opinion there is a need of a coordinator for HCT in every hospital or HCP; do you agree?Concerning international guidelines: What is the possible role of SSIEM, SIMD and other scientific organisations in preparing and distribution of these guidelines, and, especially, in organizing courses on IMDs and training for adult physicians?  HCT of complex diseases, e.g. IMD, is different from one disease to another, so there is a need for disease-specific guidelines.  Please comment on this proposal.

As you mentioned in your manuscript HCT is broader than medical care.  Do you agree that a close collaboration with (international) patients organizations is of utmost importance to make HCT a success?

You compared two anglo-saxon health care systems; introducing HCT in all different countries and continents will be a great challenge.

Author Response

I read with great interest your manuscript on an important topic of healthcare transition (HCT) in IEM which should be planned, standardized, periodically evaluated and of great importance, and as you showed independent of the organization and financing of the health care system, based on international guidelines.  In my opinion there is a need of a coordinator for HCT in every hospital or HCP; do you agree?Concerning international guidelines: What is the possible role of SSIEM, SIMD and other scientific organisations in preparing and distribution of these guidelines, and, especially, in organizing courses on IMDs and training for adult physicians?  HCT of complex diseases, e.g. IMD, is different from one disease to another, so there is a need for disease-specific guidelines.  Please comment on this proposal.

We added a discussion section to this manuscript which addresses the role of professional societies, the need for disease-specific guidelines, and the importance of HCT coordinators.

As you mentioned in your manuscript HCT is broader than medical care.  Do you agree that a close collaboration with (international) patients organizations is of utmost importance to make HCT a success?

We agree that patient support organizations should be a role in providing resources for HCT. This is noted in the discussion section.

You compared two anglo-saxon health care systems; introducing HCT in all different countries and continents will be a great challenge.

This is an astute observation of the main limitation of our report. The discussion now addresses this limitation and the importance of building a broad international coalition.
